# Etiology of Cleft Lip and/or Cleft Palate in Southeastern Poland Based on Current Observational Study

**DOI:** 10.3390/jcm14165682

**Published:** 2025-08-11

**Authors:** Margareta Budner, Patrycja Korulczyk, Agnieszka Lasota

**Affiliations:** 1Department of Plastic Reconstructive and Microsurgery, Medical University of Lublin, 20-093 Lublin, Poland; 2Medical Simulation Center, Medical University of Lublin, 20-093 Lublin, Poland; patrycjakorulczyk@umlub.pl; 3Department of Jaw Orthopedics, Medical University of Lublin, 20-093 Lublin, Poland; agnieszkalasota@umlub.pl

**Keywords:** orofacial clefts, cleft lip, cleft palate, risk factors, folic acid, tobacco

## Abstract

**Background/Objectives:** There is a need to search for risk factors of CL/P, which is the most common facial congenital deformity due to its multifactorial etiology. This study aimed to assess the possible external factors influencing the occurrence of non-syndromic CL/P. **Methods:** Retrospective–prospective case record analysis was performed with a sample of 224 consecutively treated patients. The data were obtained from medical records and questionnaires completed by children’s parents. **Results:** Factors that may have potentially increased the risk of CL/P were as follows: advanced paternal age (*p* = 0.014), mother experiencing an infection in the first trimester of pregnancy (*p* < 0.001), maternal passive smoking (*p* = 0.001) and stress during the first trimester of pregnancy (*p* = 0.003). Where mothers were treated with folic acid in the first trimester of pregnancy, in the whole sample the risk of a CL/P in newborns was reduced (*p* = 0.007). Advanced paternal age was unrelated to the occurrence of isolated CL, and stress did not increase the risk of isolated CP etiology. Maternal folic acid intake in the first trimester of pregnancy did not influence CLP occurrence. **Conclusions:** In a group of children from Southeastern Poland, the occurrence of CL/P was related to advanced maternal age and maternal external risk factors, such as infections, passive tobacco smoking and severe stress during the first trimester of pregnancy. Maternal folic acid intake during the first trimester reduced the risk of CL and CP. Due to the observational nature of the study, the above conclusions should be interpreted as a potential effect.

## 1. Introduction

The most common dentofacial non-syndromic congenital malformation in the craniofacial region is cleft lip and/or cleft palate (CL/P), with an incidence of 1 per 700 live births [1,2]. Disturbances in fusion of the medial nasal and maxillary processes between the fourth and sixth weeks of embryogenesis is a causative factor of primary palate closing failure and leads to cleft lip formation. Subsequently, lack of proper fusion of maxillary palatine processes between the seventh and twelfth weeks results in cleft palate [3]. Isolated cleft palate differs from cleft lip and cleft lip and palate in its time of formation and epidemiological features. This could indicate distinct developmental mechanisms and genetic origins of the two subtypes [4,5,6].

The classification of this anomaly is also based on its embryological origin and the structures affected. Two main types of the congenital malformation can be distinguished, the first being cleft lip without cleft palate (CL) or with cleft palate (CLP), and the second being cleft palate (CP) [7,8,9]. Due to international variation in classification systems, in 2023 the use of the LAHSHAL method was recommended for orofacial clefts [10]. CL/P is often associated with additional congenital abnormalities in various syndromes, but the more common form is non-syndromic, and occurs as an isolated condition [11]. The etiology of CL/P is attributed to genetic and environmental factors. Genetically, a complex etiology of about 40 loci has been reported, and several genetic variants associated with oral clefts have been identified in genome-wide association (GWA) studies [12,13,14,15]. The lack of a global network of registries impedes the discovery and monitoring of risk factors. Some risk factors, such as genetic polymorphisms, the gender of the affected child, ethnic origin and family history, are immutable. However, consanguinity, parental age at conception, socio-economic and educational levels and maternal weight, nutritional state, diseases, stress, intake of medication, alcohol use and smoking may be mediated by prospective parents. Accurately reporting cases of children born with clefts enables the identification of new risks such as prenatal maternal opioid exposure [16]. The need to conduct epidemiological studies specific to certain populations has been raised. With the material available from our region where cleft palate occurs quite frequently, it seems necessary to make this information available to a wider audience so that organizations and interested researchers can analyze the information and use it in further studies.

The aim of this retrospective–prospective case record analysis was the assessment of etiological risk factors of non-syndromic CL/P in Southeastern Poland.

## 2. Materials and Methods

The study group (SG) included 224 patients, comprising 103 girls and 121 boys with non-syndromic CL/P treated at the Clinic for Congenital Craniofacial Deformities, University Dental Centre, Lublin, Poland, between 2017 and 2022. All survey participants were residents of Southeastern Poland. Patient age ranged from 1 month to 16 years. The study used the documentation of all available patients. The comparison group (CG) consisted of 51 children (29 girls and 22 boys) from the Eastern Centre for Burns and Reconstructive Surgery who had experienced no facial deformities that required surgical treatment. The children recruited to the control group represented the same region of origin and the same cross-section of society. As both clinics are publicly accessible facilities and offer reimbursed treatment, they attract patients with similar socioeconomic status.

The data used in this study were obtained from a retrospective examination of patients’ medical history and questionnaires completed by the children’s parents. The researcher-administered and semi-structured questionnaire had been used and tested over a long period of time (about 25 years) as part of medical history and did not cause any problems during completion. All research protocols were approved by the Bioethical Committee of the Medical University of Lublin (nr KB-0024/148/11/2024).

Two researchers (MB and AL) documented the type of cleft, the side of the cleft in the case of unilateral CL and CLP and patients’ gender and age. In the questionnaires, parents were asked the maternal and paternal age at the time of their child’s birth, if the mother experienced infection (viral and bacterial) in the first trimester of pregnancy, the mother’s folic acid intake in the 3 months before the planned pregnancy, the mother’s folic acid intake in the first trimester of pregnancy, maternal tobacco smoking during pregnancy (including in the first trimester of pregnancy), passive maternal smoking (the mother staying at home or being at work in an environment polluted by tobacco smoke), maternal alcohol usage during the first trimester of pregnancy, maternal stress during the first trimester of pregnancy (serious illness, death of a close relative, accident, etc.), maternal educational level (vocational, secondary or higher education) and paternal educational level (vocational, secondary or higher education).

Data were inputted into a spreadsheet (Microsoft Excel Version 16.24; 2019; Microsoft, Redmond, WA, USA). Using Statistica 13.1, descriptive statistics were provided and univariate and multivariate logistic regression were performed (ss *p* < 0.05).

## 3. Results

In this study, CLP was the most common anomaly, at 41%, followed by CP, at 38%. CL was noted in 47 patients (21%). Analyzing the gender distribution, there was a statistically higher number of girls in the CP group (*p* = 0.002) while in the CLP group, boys were predominant (*p* < 0.001) (Table 1). There were no statistically significant differences in gender incidence in the CL group.

CL most frequently occurred on the left side (in 31 children—15 girls and 16 boys), less frequently on the right side (in 11 children—6 girls and 5 boys) and least frequently bilaterally (in 5 cases—3 girls and 2 boys). Moreover, CLP most frequently affected the left side (in 52 children—13 girls and 39 boys), was less frequently bilateral (in 21 children—12 girls and 9 boys) and was least frequently right-sided (in 18 children—3 girls and 15 boys). Statistically significant differences in cleft side and gender were identified in children with CLP (Table 2).

After a preliminary analysis of the environmental etiological factors of CL/P, the following were excluded from the study: maternal alcohol consumption during pregnancy (too few reported cases) and the parents’ education status (lack of statistical significance).

Multivariate analysis of other elements revealed the existence of four statistically significant CL/P risk factors and one factor that reduced risk. Factors that increased the risk were advanced paternal age, mothers experiencing infections in the first trimester of pregnancy, passive smoking (cigarette smoking by the pregnant woman’s household members during the first trimester of pregnancy) and stress during the first trimester of pregnancy (Table 3).

Based on the results, it can be concluded that increasing the age of the father by one year increased the risk of CL/P occurrence in the child 1.2-fold. Women who experienced an infection during the first trimester of pregnancy had a 12 times greater risk of having a baby with a cleft lip and/or palate compared to women who had no such infection. This same risk was eight times higher in cases where the mother experienced severe stress in the first trimester of pregnancy. Passive maternal tobacco smoking incurred a 10-fold increased risk factor for CL/P incidence. Administering folic acid in the first trimester of pregnancy reduced the risk of a CL/P in a newborn. (OR < 1)

There was no relationship between either folic acid intake before pregnancy or maternal tobacco smoking during pregnancy and CL/P incidence (Table 4).

Multivariate analysis revealed four environmental factors significantly affecting the occurrence of CL in children (Table 5). Past infections during the first trimester of pregnancy increased the risk of CL 22-fold. Maternal household smoking during the first trimester of pregnancy increased the risk 9-fold, whereas stress incurred a 13-fold increase. In contrast, the use of folic acid during the first trimester of pregnancy reduced the risk of having a child with CL (OR < 1) (Table 5).

Statistical analysis revealed three significant environmental factors that increased the risk of having a child with CP (Table 6). Increasing the father’s age by one year increased the risk of CP 1.4-fold. Women who experienced an infection had an approximately 58 times higher risk of having a child with CP compared to women who had not experienced an infection. The risk of CP occurring in the child of a mother taking folic acid during the first trimester of pregnancy was 0.04 lower than for those who did not take folic acid at that time. Women exposed to passive smoking had about a 49 times higher risk of giving birth to a child with CP compared to unexposed women. The use of folic acid during the first trimester of pregnancy reduced the risk of having a child with CP.

Multivariate analysis detected four environmental factors that increased the risk of CLP (Table 7). The age of the father on the day of the child’s birth was a statistically significant factor. Increasing the father’s age by one year increased the risk of CLP 1.2-fold. Experiencing an infection during the first trimester of pregnancy increased the risk of CLP approximately 11 times compared to that for women who did not have an infection during this time. Mothers of children who were exposed to passive tobacco smoking during the first trimester of pregnancy had an approximately 9 times higher risk of CLP. Stress in the first trimester of pregnancy increased the risk of CLP about 12-fold. The obtained results did not reveal the preventive influence of folic acid intake during the first trimester of pregnancy.

## 4. Discussion

The multifactorial etiology of CL/P is well known and widely discussed. Although the genetic findings of cleft etiology allow for a better understanding of the inheritance of CL/P, the potentially modifiable factors are of chief interest for future parents. Many different risk factors have previously been analyzed in the literature: In a Chinese population surveyed by Hong Y et al., maternal pesticide exposure and the use of antibiotic drugs were identified as risk factors of CL/P [17]. In contrast, in this study, a mother’s moderate physical workload and vitamin B complex, calcium and iron supplementation were associated with reduced risk of anomalies. In their meta-analysis, Ács et al. revealed that obesity, maternal underweight, maternal type 1 diabetes and essential hypertension were associated with higher odds of a child developing a cleft. In our study, our focus was entirely placed on the few best-proven modifiable external factors [18].

Advanced parental age was investigated as a risk factor in many studies. In Ireland, Mc Donell et al. uncovered no relationship between the rising trend in the proportion of mothers aged 35 years or older and incidences of orofacial clefts associated with a chromosomal syndrome [19]. This is in accordance with our report on non-syndromic clefts. Our study results highlight how parental age increased the risk of having a child with CL/P 1.2 times for each year of the father’s age. In a population-based study from Norway from 1967 to 2010, the analysis suggested that the risk of fathering an infant with CL increases with advancing age. However, it was discovered that the risk was increased only when the age of both parents was high [20]. Ahmed et al. conducted a case–control study of 600 CL/P cases out of Gansu Province, China, proving that the incidence of anomalies was reduced for parents aged between 25 and 29 [21]. An older study conducted by Danish researchers agrees with our study’s finding that paternal age plays an important role in the etiology of CP [22]. In this research work, advanced age in both parents was connected with the incidence of CL/P. The findings of a meta-analysis by Herkrath et al. suggested that fathers forty years of age or older had a 58% higher risk of having a child with CP compared to those aged between 20 and 39 years. In the same study, the influence of maternal age, which increased CL/P incidence, was also revealed [23].

Maternal smoking was identified as a modifiable environmental factor, which was considered a causal factor for CL/P in the 2014 US Surgeon General’s Report [24]. Also, a meta-analysis performed by Fell et al. suggested that maternal smoking might play a moderate role in CL/P etiology with a pooled OR of 1.42 (95% CI: 1.27, 1.59) [25]. In our study, maternal tobacco smoking did not influence the incidence of CL/P, perhaps because of the failure of the parents to report it. A population-based case–control study in 10 US states confirmed smoking during the month before pregnancy or the first month of pregnancy as a risk factor, with a 4.0% increase for CL/P and 3.4% increase for CP, which is lower than in our findings [26]. Reviews by other authors also showed that maternal smoking and passive smoking increase the risk of orofacial clefts [21,26,27,28,29,30,31]. Ács et al. highlighted that passive smoking was even more harmful than active tobacco smoking [18]. A meta-analysis conducted by Sabbagh et al. proved that maternal passive smoking exposure resulted in a 1.5-fold increase in the risk of non-syndromic orofacial clefts, similar to the magnitude of risk reported for active smoking [28]. According to our results and those of other authors, there is strong evidence in favor of avoiding active and passive maternal tobacco smoking before and during pregnancy.

Folic acid intake during the preconception period and the first trimester of pregnancy plays an important preventive role in the etiology of congenital deformities [32]. Our study is in agreement with other studies in that we mutually conclude that increasing the maternal folic acid intake during the first three months of pregnancy reduces the risk of CL/P. Darjazini et al. recommend the maternal intake of folic acid supplements at a dose of 0.4 to 0.8 mg during the initial trimester of pregnancy [33].

Severe maternal stress during the first three months of pregnancy was the primary CL and CLP risk factor in our study. Based on 12 studies, Talal et al. performed a meta-analysis, also proving the effect of mothers’ stress during the periconceptional period on CL/P formation [34].

Analyzing external risk factors is crucial to preventing congenital anomalies and can help in the prevention of both syndromic and non-syndromic CL/P. Not to be overlooked, the health status of children with syndromic CL/P is often more complicated, with the most frequently associated congenital anomaly being congenital heart disease [35]. In addition, an increasing tendency toward the early detection of congenital deformities is encouraging [36].

The differences in our results between cleft subtypes may be caused by distinct molecular pathways and the timing of cleft formation, as was highlighted by previous authors [4,5,6,37]

In terms of the limitations of our study, the small number of patients involved, as well as the limited number of factors analyzed, could influence the relevance of the results. It cannot be ruled out that the control group size affects the results obtained—it could be connected with the chance of false-negative or -positive effects. This leads to difficulties with generalization of the findings.

In addition, relying on a parent-completed survey might have introduced a risk of bias. For better results parents were given an explanation of the significance of the study and usually were personally interested in assessing the risk to their next offspring so it can be assumed that their answers were honest. It is difficult to obtain data for this type of study other than through a questionnaire. Unfortunately, answers to some questions such as maternal alcohol consumption or tobacco smoking may be subject to the risk of false responses. Also, control group recruitment may have been a source of the bias due to differences compared to the general population. The same residential area and similar clinic availability could improve the quality of the result

## 5. Conclusions

An advanced paternal age, the mother experiencing infection or severe stress in the first trimester of pregnancy, and passive maternal tobacco smoking may potentially increase the risk of CL/P in children. Paternal age was unrelated to isolated CL incidence. Severe maternal stress did not increase the incidence of CP. For CL and CP formation, administering folic acid in the first trimester of pregnancy displayed a preventative effect. Due to the observational nature of this study, the above conclusions should not be interpreted as a cause–potential effect result. These findings can help develop primary prevention strategies for CL/P.

## Figures and Tables

**Table 1 jcm-14-05682-t001:** Characteristics of the study group by gender and cleft type.

	CL	CP	CLP	
Gender				
Girls	24 (10.71%)	51 (22.77%)	28 (12.50%)	103 (45.98%)
Boys	23 (10.27%)	35 (15.63%)	63 (28.13%)	121 (54.02%)
Total	47 (20.98%)	86 (38.39%)	91 (40.63%)	224 (100.00%)

**Table 2 jcm-14-05682-t002:** Cleft side and type in the study group.

Side and Type of the Cleft	Gender		*p* **
	Girls	Boys	
CL, left side	15 (28.85%)	16 (18.60%)	0.162
CL, right side	6 (11.54%)	5 (5.81%)	0.228
CL, bilateral	3 (5.77%)	2 (2.33%)	0.295
CLP, left side	13 (25.00%)	39 (45.35%)	**0.017**
CLP, right side	3 (5.77%)	15 (17.44%)	**0.049**
CP, bilateral	12 (23.08%)	9 (10.47%)	**0.046**
**Total**	52 (100.00%)	86 (100.00%)	-
**χ^2^; *p* ***	14.239; 0.014		

* *p*—significance level for the χ^2^. ** *p*—significance level for the difference test between two structure indicators.

**Table 3 jcm-14-05682-t003:** Results of estimation model describing CL/P incidence.

Risk Factor	B	SE	χ^2^ Wald	*p*	OR (95% CI)
Maternal age	−0.054	0.079	0.463	0.496	0.948 (0.812–1.106)
Paternal age	0.182	0.074	6.095	**0.014**	1.200 (1.038–1.387)
Infections	2.494	0.702	12.623	**<0.001**	12.107 (3.059–47.916)
Folic acid intake before pregnancy	−0.092	0.494	0.035	0.852	0.912 (0.346–2.401)
Folic acid intake during pregnancy	−1.673	0.623	7.208	**0.007**	0.188 (0.055–0.637)
Maternal tobacco smoking during pregnancy	1.219	1.168	1.168	0.297	3.383 (0.343–33.372)
Passive maternal tobacco smoking during pregnancy	2.255	0.703	10.278	**0.001**	9.533 (2.402–37.834)
Stress during pregnancy	2.124	0.720	8.702	**0.003**	8.361 (2.039–34.280)

B—evaluation of model parameters. SE—standard error. χ^2^ Wald—value of the χ^2^ statistic checking the significance of the parameters. *p*—significance level for the Wald test. OR (95% CI)—odds ratio and 95% confidence intervals.

**Table 4 jcm-14-05682-t004:** Descriptive statistics and comparison of factors influencing the occurrence of cleft lip and/or palate.

Risk Factor	SG		CG		*p*
	*n*	%	*n*	%	
Infections	3	5.88	53	28.49	**0.001**
Acid folic intake before pregnancy	15	29.41	31	16.49	0.061
Acid folic intake during pregnancy	45	90.00	132	70.21	**0.008**
Maternal tobacco smoking during pregnancy	1	1.96	19	10.16	0.113
Maternal tobacco passive smoking during pregnancy	3	5.88	80	42.78	**<0.001**
Maternal alcohol consumption	0	0.00	3	1.60	-
Stress during pregnancy	3	5.88	39	22.67	**<0.001**

*p*—significance level for the test of differences between two structural indicators.

**Table 5 jcm-14-05682-t005:** Results of the estimation model describing the CL incidence.

Risk Factor	B	SE	χ^2^ Wald	*p*	OR (95% CI)
Maternal age	−0.051	0.114	0.205	0.651	0.950 (0.760–1.187)
Paternal age	0.166	0.094	3.123	0.077	1.181 (0.982–1.420)
Infections	3.071	0.983	9.761	**0.002**	21.573 (3.141–148.162)
Folic acid intake before pregnancy	0.194	0.753	0.066	0.797	1.214 (0.277–5.312)
Folic acid intake during pregnancy	−2.232	0.868	6.611	**0.010**	0.107 (0.020–0.588)
Maternal tobacco smoking during pregnancy	1.875	1.588	1.394	0.238	6.522 (0.290–146.699)
Passive maternal tobacco smoking during pregnancy	2.217	0.881	6.331	**0.012**	9.180 (1.633–51.626)
Stress during pregnancy	2.528	0.962	6.902	**0.009**	12.523 (1.900–82.533)

B—evaluation of model parameters. SE—standard error. χ^2^ Wald—value of the χ^2^ statistic checking the significance of the parameters. *p*—significance level for the Wald test. OR (95% CI)—odds ratio and 95% confidence intervals.

**Table 6 jcm-14-05682-t006:** Results of estimation model describing CP incidence.

Risk Factor	B	SE	χ^2^ Wald	*p*	OR (95% CI)
Maternal age	−0.094	0.106	0.794	0.373	0.910 (0.739–1.120)
Paternal age	0.366	0.124	8.723	**0.003**	1.443 (1.131–1.840)
Infections	4.061	1.106	13.468	**<0.001**	58.004 (6.632–507.303)
Folic acid intake before pregnancy	−1.238	0.813	2.320	0.128	0.290 (0.059–1.426)
Folic acid intake during pregnancy	−3.113	1.030	9.142	**0.002**	0.044 (0.006–0.335)
Maternal tobacco smoking during pregnancy	2.066	1.293	2.551	0.110	7.891 (0.625–99.575)
Passive maternal tobacco smoking during pregnancy	3.882	1.212	10.269	**0.001**	48.539 (4.517–521.590)
Stress during pregnancy	3.102	1.087	8.146	**0.004**	22.249 (2.643–187.282)

B—estimation of model parameters. SE—standard error. χ^2^ Wald—value of the χ^2^ statistic checking the significance of the parameters. *p*—significance level for the Wald test. OR (95% CI)—odds ratio and 95% confidence intervals.

**Table 7 jcm-14-05682-t007:** Results of the estimation model describing CLP incidence.

Risk Factor	B	SE	χ^2^ Wald	*p*	OR (95% CI)
Maternal age	−0.057	0.101	0.316	0.574	0.945 (0.775–1.152)
Paternal age	0.197	0.098	4.049	**0.044**	1.217 (1.005–1.474)
Infections	2.357	0.842	7.832	**0.005**	10.556 (2.026–54.993)
Folic acid intake before pregnancy	0.378	0.622	0.370	0.543	1.460 (0.431–4.938)
Folic acid intake during pregnancy	−1.102	0.739	2.223	0.136	0.332 (0.078–1.415)
Maternal tobacco smoking during pregnancy	−0.355	1.975	0.032	0.857	0.701 (0.015–33.674)
Passive maternal tobacco smoking during pregnancy	2.193	0.759	8.346	**0.004**	8.963 (2.024–39.684)
Stress during pregnancy	2.457	0.892	7.583	**0.006**	11.667 (2.030–67.042)

B—evaluation of model parameters. SE—standard error. χ^2^ Wald—value of the χ^2^ statistic checking the significance of the parameters. *p*—significance level for the Wald test. OR (95% CI)—odds ratio and 95% confidence intervals.

## Data Availability

Data are available from corresponding author upon reasonable request.

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
