# Peer review of "Etiology of Cleft Lip and/or Cleft Palate in Southeastern Poland Based on Current Observational Study"

_jcm, 2025, doi:10.3390/jcm14165682_

Round 1
Reviewer 1 Report
Comments and Suggestions for Authors
The manuscript titled “Contemporary Etiology of Cleft Lip and/or Cleft Palate in Southern Eastern Poland” was submitted to JCM.
The authors conducted a retrospective-prospective case record analysis of 224 patients with nonsyndromic cleft lip and/or palate (CL/P) treated between 2017 and 2022 in Southern Eastern Poland. Data were collected via medical records and parental questionnaires to identify external etiological risk factors. The comparison group included 51 children without facial deformities. Multivariate logistic regression identified several statistically significant risk factors: increased paternal age, maternal infection, passive tobacco exposure, and stress during the first trimester of pregnancy. Folic acid intake during early pregnancy was identified as a protective factor for cleft lip and cleft palate but not for CLP. The study concludes that modifiable maternal risk factors and paternal age contribute significantly to nonsyndromic CL/P risk.
Although the study presents an important topic in craniofacial anomaly research and offers original data from a specific geographic region, several significant some corrections need to be made.
Specific comments are outlined below.
Title
The title lacks clarity regarding the study design, which should be explicitly stated
Moreover, the phrasing "Contemporary etiology" is vague and could be better framed.
Abstract
Several grammatical errors impair clarity and flow (e.g., “Folic acid mothers’ intake...” should be “Maternal folic acid intake…”).
The statistical results are not uniformly presented; for example, odds ratios should include confidence intervals consistently.
Keywords: Make sure they correspond to MeSH terms.
Introduction
The introduction provides an overview of cleft classifications and genetic causes but lacks a well-defined problem statement and rationale for focusing on environmental factors in the Polish population.
Besides, several references are outdated or general; the inclusion of more recent epidemiological or population-specific data would strengthen the justification.
Materials and Methods
There is no clarity about the study design
Besides, there is ambiguity in the description of the comparison group. The source (Burns and Reconstructive Surgery center) may not be a demographically matched control population, possibly introducing selection bias.
Furthermore, the methodology does not clearly state how cases and controls were matched or adjusted for confounding variables (e.g., socioeconomic status, urban/rural setting).
Also, the data collection tool (parent questionnaire) is not validated or described in sufficient detail.
Ethical approval is listed, but there is no statement on informed consent.
Besides, the statistical methodology is underdeveloped. No rationale is provided for the choice of statistical tests, and the handling of missing data is not explained.
There is a paragraph of editorial instructions in the middle of the section (lines 75–94) that should have been removed before submission.
Results
Tables are overloaded with data, and results are sometimes repeated in narrative form without synthesis.
Moreover, the distinction between statistical significance and clinical relevance is not discussed.
Some claims in the results section are confusingly worded (e.g., “reduced the risk by 0.19 times” should be reframed using ORs < 1.0).
Also, tables lack clarity in layout and consistent use of decimal places and formatting (e.g., using commas for decimals instead of periods).
Discussion
The discussion largely reiterates findings rather than providing deep interpretation.
Moreover, there is limited discussion on limitations—only a brief mention of small sample size and questionnaire bias at the end. More attention should be given to the lack of control for confounding, recall bias, and generalizability.
The literature review lacks integration; it reads like a list of findings rather than a synthesis.
Also, several references cited (e.g., [17], [24], [27]) do not seem to be critically compared to the current findings.
Some statements are speculative or not well-supported by the data (e.g., assumptions about parental misreporting of smoking).
Conclusion
The conclusions are too definitive given the limitations of the observational study design.
Moreover, the sentence “Fathers’ age had no relation with CL incidence” contradicts earlier results without adequate explanation.
There is no distinction between hypothesis-generating findings and those robust enough to guide clinical practice.
References
Several are outdated (e.g., reference 2 from WHO dated 2001).
Author Response
Thank you very much for your insightful and professional review. The reviewer's comments allow to learn how to plan and write a scientific paper correctly.
Comments 1: Title. The title lacks clarity regarding the study design, which should be explicitly stated
Moreover, the phrasing "Contemporary etiology" is vague and could be better framed.
Response1: The title was changed due to your recommendations
Comments 2: Abstract: Several grammatical errors impair clarity and flow (e.g., “Folic acid mothers’ intake...” should be “Maternal folic acid intake…”). The statistical results are not uniformly presented; for example, odds ratios should include confidence intervals consistently.
Keywords: Make sure they correspond to MeSH terms.
Response 2. The grammatical errors have been corrected, statistical results were standardised by p- value insertion( OR with CI is included in the tables), changes were performed in key words for better corresponding with MeSH terms
Comments 3: Introduction. The introduction provides an overview of cleft classifications and genetic causes but lacks a well-defined problem statement and rationale for focusing on environmental factors in the Polish population. Besides, several references are outdated or general; the inclusion of more recent epidemiological or population-specific data would strengthen the justification.
Response 3: The aim of the study was better explained and more recent epidemiological reference was added. Unfortunately there is a lack of new epidemiological studies in Poland.
Comments 4: Materials and Methods There is no clarity about the study design. Besides, there is ambiguity in the description of the comparison group. The source (Burns and Reconstructive Surgery center) may not be a demographically matched control population, possibly introducing selection bias. Furthermore, the methodology does not clearly state how cases and controls were matched or adjusted for confounding variables (e.g., socioeconomic status, urban/rural setting). Also, the data collection tool (parent questionnaire) is not validated or described in sufficient detail. Ethical approval is listed, but there is no statement on informed consent. Besides, the statistical methodology is underdeveloped. No rationale is provided for the choice of statistical tests, and the handling of missing data is not explained. There is a paragraph of editorial instructions in the middle of the section (lines 75–94) that should have been removed before submission.
Response 4: The study design is observational based on patients records. We used existing records and also after decission about the study we collected data from newly arriving patients and sometimes had to fill in the missing information. The control group recruitment was better described in the text according to your advice. Used questionnaires was researcher-administered and semi-structured. As it has been using for many (25) years it has been tested and revised many times an during our study it did not cause any difficulties. The proper description was added to the text. The statement on informed consenent is included in the end of manuscript.
Comments 5: Results. Tables are overloaded with data, and results are sometimes repeated in narrative form without synthesis.
Moreover, the distinction between statistical significance and clinical relevance is not discussed. Some claims in the results section are confusingly worded (e.g., “reduced the risk by 0.19 times” should be reframed using ORs < 1.0). Also, tables lack clarity in layout and consistent use of decimal places and formatting (e.g., using commas for decimals instead of periods).
Response 5: The data presented in the table seemed valuable to the readers intersting in the examined subject and we met some difficulties in summarise them. Also some repetition of infomation in the text and in the tables emphasised the problem. We changed some phrases which were not clear and replace commas.
Comments 6: Discussion The discussion largely reiterates findings rather than providing deep interpretation. Moreover, there is limited discussion on limitations—only a brief mention of small sample size and questionnaire bias at the end. More attention should be given to the lack of control for confounding, recall bias, and generalizability. The literature review lacks integration; it reads like a list of findings rather than a synthesis. Also, several references cited (e.g., [17], [24], [27]) do not seem to be critically compared to the current findings. Some statements are speculative or not well-supported by the data (e.g., assumptions about parental misreporting of smoking).
Response 6: In the discussion part we analyzed our findings in the relation to the literature which was chosen due to number of citation and publication time. Due to recommendations the section of the discussion containing limitations was developed and also some inappropriate statements were explained.
Comments: 7 Conclusion. The conclusions are too definitive given the limitations of the observational study design.
Moreover, the sentence “Fathers’ age had no relation with CL incidence” contradicts earlier results without adequate explanation. There is no distinction between hypothesis-generating findings and those robust enough to guide clinical practice.
Response 7: The conclusions were modified due to recommendations.

Reviewer 2 Report
Comments and Suggestions for Authors
Dear authors, I find your manuscript interesting, but some concerns must be raised. The sample size analysis is missing, as is a description of the statistics used. Rows 75-94 must be deleted. In the Introduction section, rows 19 and 22 are in contradiction. It is not clear if the mothers' folic intake influences CLP.
Author Response
Thank You very much for your revision.
Comments 1. The sample size analysis is missing, as is a description of the statistics used:
Response For the sample all available patients were recruited. All exclusions were due to failure to meet the inclusion criteria for the study. The description of the statistical method is in th last row of Material and Methods chapter (highlighted with yellow)
Rows 75-94 must be deleted. - It was deleted.
In the Introduction section, rows 19 and 22 are in contradiction. It is not clear if the mothers' folic intake influences CLP:
Maybe misunderstanding is due to using abbreviation: Considering the whole sample: CL, CP and CLP (abbreviation for all groups CL/P) the mother's folic intake reduced the risk of cleft. Considering subphentopes the stuation was different : in the full cleft lip and palate (CLP) folic acid intake in the first trimester of pregnancy did not influence cleft occurrence.

Reviewer 3 Report
Comments and Suggestions for Authors
This is a valuable and well-structured study that investigates environmental risk factors associated with nonsyndromic cleft lip and/or palate (CL/P) in a specific population in Southeastern Poland. The case-control design, clear subgroup analysis (CL, CP, CLP), and inclusion of multivariate logistic regression enhance the scientific rigor of the work.
However, the manuscript would benefit from the following improvements:
- Potential Selection and Recall Bias
The study design relies on retrospective questionnaires filled out by parents, which may be prone to recall bias, especially for sensitive topics like stress, smoking, and alcohol use during pregnancy. Additionally, there could be selection bias since the control group was sourced from a burn and reconstructive surgery center, which may not accurately represent the general population. Please explicitly discuss these limitations. - Statistical Power Considerations
The control group consists of only 51 individuals, potentially limiting the ability to detect significant differences in some comparisons (e.g., maternal active smoking and alcohol consumption). Please address how the small control group size might impact the robustness of the results. - Adjustment for Confounding Variables
Although multivariate logistic regression was used, it is unclear whether the models accounted for potential confounders such as socioeconomic status, maternal education, or collinearity between maternal and paternal age. Consider discussing these factors and whether additional adjustments or stratified analyses were considered. - Definition and Objectivity of Risk Factors
The categorization of exposures like “maternal stress,” “infection,” or “passive smoking” lacks clear, standardized definitions. For example, the specific types of infections (viral vs. bacterial), the nature and severity of stress, or the measurement of secondhand smoke exposure are not described in detail. These limitations could impact the reproducibility and interpretation of the reported associations. - Biological Mechanisms and Interpretation by Subgroup (CL, CP, CLP)
The subgroup analyses reveal different patterns of association between environmental factors and each cleft subtype (CL, CP, CLP). However, the biological plausibility and developmental mechanisms behind these subtype-specific differences are not thoroughly discussed. Including references to embryological or molecular pathways could strengthen the interpretation. - Causality vs. Association
Since the study is observational, it cannot establish causal relationships definitively. Phrases like “increased the risk by 10 times” may be misinterpreted as causal. Consider revising such language to emphasize that the study identifies associations, not causation.
English Language and Grammar:
The manuscript requires a thorough revision of grammar, syntax, and vocabulary. Multiple grammatical and typographical errors (e.g., "did not increased", "billateral", inconsistent use of commas in p-values) affect clarity.
It is strongly recommended that the authors seek editing by a native English speaker or professional language editing service.
Terminology Consistency:
Terms such as "folic acid intake" are inconsistently expressed as "acid folic intake" or "folic acid mothers’ intake". Please standardize terminology throughout the text.
Presentation of Results:
The reporting of odds ratios is appropriate; however, clarify whether "reduced risk by 0.19 times" means a reduction of 81% (i.e., OR=0.19). Using percentage reduction consistently would improve readability.
Ensure p-values are presented with consistent decimal notation (e.g., “p = 0.002” instead of “p = 0,002”).
Minor Points:
Clarify ambiguous or imprecise phrases, such as “risk factor of increasing 10 times CL/P incidence” — rephrase for clarity.
Spell-check all technical terms, such as “billateral” → “bilateral”.
Despite these issues, the conclusions are generally well supported by the data, and the study contributes meaningfully to the understanding of modifiable risk factors for CL/P. I encourage the authors to revise the manuscript accordingly.
Comments on the Quality of English LanguageThe manuscript is generally understandable; however, the quality of English needs moderate to major revisions. There are multiple issues with grammar, syntax, article use, and vocabulary throughout the text. For example, inconsistent use of articles, verb tense errors (such as “did not increased” instead of “did not increase”), and awkward phrasing (like “Folic acid mothers’ intake”) reduce clarity. Additionally, some typographical errors, such as misspelling “bilateral” as “billateral” and using commas instead of periods in p-values (e.g., “p=0,002”), should be corrected.
A thorough revision by a native English speaker or professional editing service is strongly recommended to enhance the readability and overall presentation of the manuscript.
Author Response
We are very grateful for your big effort in analysisng our manuscript. We hope our revision will meet your recomendation.
Comments 1: Potential Selection and Recall Bias. The study design relies on retrospective questionnaires filled out by parents, which may be prone to recall bias, especially for sensitive topics like stress, smoking, and alcohol use during pregnancy. Additionally, there could be selection bias since the control group was sourced from a burn and reconstructive surgery center, which may not accurately represent the general population. Please explicitly discuss these limitations.
Response 1 : According to your comment these limitations were added to the Discussion part.
Comments 2: Statistical Power Considerations. The control group consists of only 51 individuals, potentially limiting the ability to detect significant differences in some comparisons (e.g., maternal active smoking and alcohol consumption). Please address how the small control group size might impact the robustness of the results.
Response 2. The control group size could influence the results by the risk of false negatives or positives effects. Of course it could also leads to difficulties in generalization. This issue was added to the limitations of the study.
Comments 3. Adjustment for Confounding Variables . Although multivariate logistic regression was used, it is unclear whether the models accounted for potential confounders such as socioeconomic status, maternal education, or collinearity between maternal and paternal age. Consider discussing these factors and whether additional adjustments or stratified analyses were considered.
Response 3. In the preliminary calculations we checked the significance of all examined factors and for final regression models we used only those with the influence. All factors like: socio-economic status, maternal or paternal educational level were excluded. We didn't performed stratified analysys, although it is interesting consideration for futher research.
Comments 4: Definition and Objectivity of Risk Factors. The categorization of exposures like “maternal stress,” “infection,” or “passive smoking” lacks clear, standardized definitions. For example, the specific types of infections (viral vs. bacterial), the nature and severity of stress, or the measurement of secondhand smoke exposure are not described in detail. These limitations could impact the reproducibility and interpretation of the reported associations.
Response 4: All explanations of the used terms were included to the text as a revision suggested.
Comments 5: Biological Mechanisms and Interpretation by Subgroup (CL, CP, CLP) The subgroup analyses reveal different patterns of association between environmental factors and each cleft subtype (CL, CP, CLP). However, the biological plausibility and developmental mechanisms behind these subtype-specific differences are not thoroughly discussed. Including references to embryological or molecular pathways could strengthen the interpretation.
Response 5: Proposed information were added to Introduction and Discussion part together with references.
Comments 6: Causality vs. Association. Since the study is observational, it cannot establish causal relationships definitively. Phrases like “increased the risk by 10 times” may be misinterpreted as causal. Consider revising such language to emphasize that the study identifies associations, not causation.
Response 6: As recommended in the review, we described the limitations of the observational study. We added explanations to the abstract, conclusions and methods. The terms used are abbreviations used in other articles in the literature, and the shortcomings result from the use of English language rather than the original language.
English Language and Grammar: The manuscript was revised by MDPI services, Certficate is added in the attachement
Terminology Consistency: the terminology was standardized
Presentation of Results: In reduction percentage is readable, but the problem is with increase, because it gives more than 100 %.
All p- values have got decimal notation.

Round 2
Reviewer 2 Report
Comments and Suggestions for Authors
The authors accepted my suggestions, and the manuscript can be publish.
Reviewer 3 Report
Comments and Suggestions for Authors
Thank you for your careful revisions. The manuscript is now clearer and better structured. I appreciate your efforts to address the previous comments and to improve the discussion.
I have no further suggestions and support the publication of your work.